# Nitric Oxide in Macrophage Immunometabolism: Hiding in Plain Sight

**DOI:** 10.3390/metabo10110429

**Published:** 2020-10-26

**Authors:** Erika M. Palmieri, Christopher McGinity, David A. Wink, Daniel W. McVicar

**Affiliations:** Laboratory of Cancer ImmunoMetabolism, National Cancer Institute, Frederick, MD 21702, USA; erikamariana.palmieri@nih.gov (E.M.P.); chris.mcginity@nih.gov (C.M.); wink@mail.nih.gov (D.A.W.)

**Keywords:** macrophage, metabolic reprogramming, nitric oxide

## Abstract

Nitric Oxide (NO) is a soluble endogenous gas with various biological functions like signaling, and working as an effector molecule or metabolic regulator. In response to inflammatory signals, immune myeloid cells, like macrophages, increase production of cytokines and NO, which is important for pathogen killing. Under these proinflammatory circumstances, called “M1”, macrophages undergo a series of metabolic changes including rewiring of their tricarboxylic acid (TCA) cycle. Here, we review findings indicating that NO, through its interaction with heme and non-heme metal containing proteins, together with components of the electron transport chain, functions not only as a regulator of cell respiration, but also a modulator of intracellular cell metabolism. Moreover, diverse effects of NO and NO-derived reactive nitrogen species (RNS) involve precise interactions with different targets depending on concentration, temporal, and spatial restrictions. Although the role of NO in macrophage reprogramming has been in evidence for some time, current models have largely minimized its importance. It has, therefore, been hiding in plain sight. A review of the chemical properties of NO, past biochemical studies, and recent publications, necessitates that mechanisms of macrophage TCA reprogramming during stimulation must be re-imagined and re-interpreted as mechanistic results of NO exposure. The revised model of metabolic rewiring we describe here incorporates many early findings regarding NO biochemistry and brings NO out of hiding and to the forefront of macrophages immunometabolism.

## 1. Introduction

For many years, nitric oxide (NO) has been appreciated as an important factor in physiology. The discovery of this endogenous radical came from studies in two scientific fields simultaneously, the cardiovascular system and the immune system. Early descriptions of an Endothelium-derived relaxing factor (EDRF) capable of inducing vasodilation began in 1980 [1]. The factor was induced by acetylcholine stimulation of endothelial cells, transferable, and unstable. It worked by stimulating soluble guanylate cyclase (sGC) and could be inhibited by hemoglobin, superoxide, and methylene blue [2]. In 1987, Robert Furchgott proposed, and Lou Ignarro showed, with HbO_2_ and chemiluminescence, that EDRF was nitric oxide mediating the activity of sGC [3,4,5]. Meanwhile, since the early 1970s John Hibbs had been studying the ability of activated macrophages to kill tumor cells, and he documented the ability of endocytosed hemoglobin to block this activity [6]. He described the mechanism of macrophage killing to involve inhibition of mitochondrial respiration. Hibbs later refined this mechanism by demonstrating inhibition of the metabolic enzyme aconitase in tumor targets, establishing the need for l-arginine for the cytotoxic effect. Stuehr and Marletta showed that macrophages produce nitrite and nitrate as well nitrosamine, indicative of nitrosation [7]. In fact, l-arginine and l-arginine deiminase activity were directly involved, and the production of nitrate and nitrite were associated with macrophage cytotoxicity [7]. In 1988–89, Hibbs as well as Stuehr and Nathan [8,9] connected the dots by showing NO to be the key effector molecule used by activated murine macrophages to kill tumors. 

These pioneering observations helped to launch a prolific field in biomedicine leading to the 1998 Nobel prize in Physiology or Medicine being awarded to Robert Furchgott, Louis Ignarro, and Ferid Murad for the role of NO in cardiovascular signaling and for *Science* to name NO the “Molecule of the Year” in 1992. Despite the rich history of NO research, its abundant production by activated macrophages, and the descriptions of macrophage effector activity being mediated via alteration of target cell mitochondrial biochemistry, NO was largely ignored during much of the recent explosion of interest in metabolic reprogramming. Although present for years hiding in plain sight, only recently has the immunometabolism world begun to embrace the importance of this simple, highly diffusible signaling molecule in macrophage metabolic programming. Moreover, with the development of compelling evidence indicating a profound role for NO in the direct control of cancer cell biology [7,10,11,12,13,14] and the known role of myeloid cells in cancer-associated immunosuppression and cancer immunotherapies, an appreciation of NO’s role in immunometabolism now takes on renewed importance for cancer biologists and immunologists alike. Herein, we review the production and the relevant chemistry of NO and integrate the early work on metabolism with more recent findings from us and others. The data demonstrate a profound role for NO in the control of macrophage metabolic reprogramming and necessarily prompts re-examination of immunometabolic work that may have failed to fully consider a possible role for NO.

## 2. Production

NO is synthesized by the NO synthase (NOS) enzymes that convert l-arginine and molecular oxygen (O_2_) to NO and l-citrulline via the intermediate metabolite n-hydroxy-l-arginine (NOHA) utilizing electrons donated by NADPH [15] (Figure 1a). Under most normal physiological conditions, constitutive nitric oxide synthases NOS1 (endothelial) and NOS3 (neuronal) generate NO at nanomolar concentrations. In these cases, NO is not rapidly oxidized and can interact directly with metal ions in proteins, for example the iron atom in the heme group of hemoglobin, myoglobin, sGC, cytochrome P450, and NOS itself [16,17,18,19]. The rate constant for association of NO to heme centers is typically very high relative to other potential targets [20], such that even low nanomolar levels of NO are sufficient to activate sGC [21]. Under these conditions, the rapid reactions between NO and targets with these favorable kinetics, limits alternative signaling processes that may otherwise be deleterious to cellular function [21]. The activation of sGC by NO yields cyclic guanosine monophosphate (cGMP), which in turn activates cGMP-dependent protein kinases mediating relaxation, neuronal transmission, and inhibition of platelet aggregation.

In contrast to vascular integrity, the roles of NO in immunity involve it being a toxic agent towards infectious organisms [13,22,23], an inducer or suppressor of leukocyte apoptosis [24] or an immunoregulator [25,26,27,28,29,30,31]. Because the immune system is activated in response to infection, an NO response from the immune system is developed over hours, days, or weeks. Moreover, for NO to be effective as a toxic or regulatory immune mediator it often needs to be generated at micromolar levels for a sustained period of time. Therefore, evidence suggests that the high NO production sustained by inducible NOS (iNOS or NOS2) is fundamental for the biology of NO linked to inflammation, in contrast to NO pulses produced by the constitutive NOS enzymes that elicit a low NO wave within fractions of seconds as needed in cardiovascular responses [15,22,26].

Even within macrophages, there is substantial variation in the NO flux derived from various stimuli. Somasundaram et al. recently detailed how different cytokines +/− lipopolysaccharide (LPS) stimulate macrophage cell lines and Bone Marrow Derived Macrophages (BMDMs) to produce different NO fluxes [32]. They find that NO flux depends on both expression of *Nos2* and on the density of cells expressing *Nos2*. They characterized levels of NO flux as being low, intermediate, or high, respectively, upon treatment with IFNγ+IL-1β, IFNγ+TNFα, and IFNγ+LPS [33]. Accordingly, they found *Nos2* induction was temporally regulated based on cytokine combinations. With strong stimuli, *Nos2* induction occurs as early as 4 h. Following real-time production of the key NO-derived reactive nitrogen species (RNS), N_2_O_3_, nitrosation potential peaks at 24 h, while extracellular nitrite levels and NOS2 protein expression peak at 42 h. Only stimulation with IFNγ+LPS strongly induces *Nos2* and NO, while other cytokine combinations are weaker stimulants and slower inducers. Moreover, Somasundaram et al. also confirmed citrulline as a robust and reliable intracellular metabolic readout and biomarker of NO flux in macrophages while arginine consumption by arginase is negligible.

The variability in the rate of production of NO from different cell types and in various stimulation conditions thus provides the necessary perspective on how NO may be important in normal physiology while also playing roles in antipathogen responses [34]. At the higher NO levels achieved during inflammatory responses, other reactions, more kinetically dependent and regulated by concentration, spatial distribution, and duration of NO flux, begin to take place along with the scavenging of radicals and interactions with heme proteins. NO can now coordinate with non-heme iron targets in the cytosol. Moreover, the conversion of NO into RNS that occurs during high NO flux can lead to indirect signaling through nitrosative, nitrative, and oxidative modifications. Such modifications involve redox reactions and depend significantly on the cellular redox landscape with respect to the presence of nitrogen, oxygen, and sulfur-related species [21].

When considering modifications, it is important to define ‘nitrosylation’ when referring to direct addition of NO to a reactant. This term has classically been used to describe the coordination of NO to a metal center to form a metal-nitrosyl complex. The coordination of NO to the ferrous heme of sGC is the traditional example of nitrosylation in biology [35]. In contrast, ‘nitrosation’ refers to reactions involving a nucleophilic group, such as an amine or thiolate, and the nitrosonium ion (NO^+^; the one-electron oxidized form of NO) yielding RN-NO or RS-NO. Nitrosation requires NO^+^ donors since in solution, NO^+^ has an appreciable lifetime only under high acidic conditions [35]. The term ‘nitrosative stress’ refers to the indiscriminate nitrosation of biological nucleophiles that can lead to pathophysiological conditions. Nitrogen oxides such as NO_2_ and peroxynitrite (ONOO^−^), although potent oxidants, cannot directly participate in nitrosation chemistry, but they can serve as precursors to nitrosating species.

## 3. NO as Competitive Inhibitor of Complex IV 

Early reports concerning the reaction of NO with cytochrome c oxidase, complex IV of the electron transport chain (ETC), date back to 1955 [36]. Cytochrome c oxidase catalyzes the oxidation of cytochrome c and the reduction of O_2_ to water in a process linked to the pumping of protons out of the mitochondrial matrix. Cytochrome c oxidase is situated on the inner membrane of the mitochondrion, and it contains two heme (a and a3) and two copper centers (Cu A and Cu B ), of which the heme iron of a3 together with Cu B, both in reduced forms, constitute the binding site for O_2_. Because NO closely resembles O_2_, like carbon monoxide (CO), it binds to the same site in the enzyme as O_2_ (Figure 1b).

In 1994, several groups found that NO inhibits cytochrome c oxidase potently and reversibly in mammalian systems [37,38,39], thereby decreasing O_2_ consumption. They found that NO reduces the affinity of the enzyme for O_2_, increasing its K_m_. In later studies, NO was found to bind and dissociate very rapidly from the enzyme [40,41]. Importantly, the inhibitory effect of NO on cytochrome c oxidase is totally inversely dependent on the concentration of O_2_. At an O_2_ concentration of 145 µM (approximately same as arterial concentration), the IC_50_ for NO inhibition is 270 nM, whereas at 30 µM (approximately the tissue concentration of O_2_) the IC_50_ falls to 60 nM [38]. Intracellularly, the concentration of O_2_ has been reported to be as low as 10 µM [42]. At this concentration of O_2_, the IC_50_ of NO for cytochrome c oxidase is predicted to be ~20 nM, suggesting that the concentrations of NO that have been detected in tissues (10–450 nM) [43,44,45] would be sufficient to effectively compete with intracellular O_2_ for cytochrome c oxidase. These conditions make cytochrome c oxidase nearly as sensitive to NO as sGC. A consequence of NO inhibition of cytochrome c oxidase would be increased availability of O_2_ in the surrounding tissues [46,47]. As reduced supplies of O_2_ will make endogenous NO even more effective at inhibiting cytochrome c oxidase [48], this might indeed be a mechanism in the microcirculation by which NO, sensing hypoxia, diverts O_2_ into adjacent issues and at the same time induces vasodilation in an attempt to improve tissue perfusion [45]. 

An extension of this concept is that, in addition to regulating cytochrome c oxidase, NO has the potential to generate a ‘metabolic hypoxia’, in situations where due to high levels of NO, available O_2_ cannot adequately be used. Such a mechanism has been proposed to occur during sepsis [49], in tissues where NO concentrations are increased, and this might also be the basis for some of the cytotoxic actions of macrophages and other phagocytic cells [50]. Indeed, recent evidence suggests the ability of NOS2 to control hypoxic gradients within the tumor microenvironment (TME). [32].

## 4. Inhibition of Non-Heme Enzymes

Reports from 1980 began to show that incubation of a variety of neoplastic cell types with cytotoxic macrophage-conditional medium (CM) leads to inhibition of target cell mitochondrial respiration by as much as 80–90% [51]. In a follow up study, the same authors investigated the nature of this inhibition by studying mitochondrial respiration in CM-injured leukemia L1210 cells. They found that Complex I (NADH-coenzyme Q reductase) was most sensitive to inhibition, followed by Complex II (succinate-coenzyme Q reductase) [52]. Cytochrome c oxidase was inhibited to a minor degree and in this system electron flow from coenzyme Q to cytochrome c was not inhibited until longer times of coculture.

Drapier and Hibbs further dissected this phenomenon and found that activated macrophages inhibit the citric acid cycle enzyme aconitase in target tumor cells [53]. In addition, cytotoxic macrophages caused a loss of iron in cocultured tumor cells. Target cell aconitase activity decreases early in the coculture and its inhibition is due to removal of iron from its iron-sulfur prosthetic group, consistent with earlier work showing that oxidative stress results in the conversion of the 4Fe-4S cluster of the active enzyme into 3Fe-4S of the inactive enzyme [54,55,56] (Figure 1c). Reducing agents or incubation of tumor cells with ferrous iron and thiosulfate or cysteine were able to restore aconitase activity in these cocultures. These authors found that the kinetics of mitochondrial aconitase (ACO2) inhibition was faster than that of the inhibition of Complex I and II of the mitochondrial ETC. Endogenous respiration continued at a normal rate with aconitase activity blocked but Complex I and II were still functional. Substantial reduction in respiration was not seen till later time points. Complex I and II may follow a similar mechanism that could explain cytotoxic activated macrophage mediated inhibition of respiration, as Complex I contains 3 4Fe-4S clusters [57] and Complex II contains a 3 iron cluster of the 3Fe-4S type [58]. The inhibitory effect would be caused by NO-mediated degradation of iron-sulfur centers resulting in the release of Fe ions and Fe-nitrosyl complexes, the first evidence of which was reported is 1983 in cells of *Clostridium Botulinum*. Through Electron Spin Resonance, iron-nitric oxide complexes were detected in proteins in these cells [59]. However, assumption and generalization that all enzymes containing Fe-S clusters are inhibited by the same mechanism cannot be made, as Complex III, although containing a 2Fe-2S cluster essential for electron transport function [57], was found to be still functional in injured target cells. It should be noted that, due to its hydrophobic nature NO significantly concentrates in lipophilic cell compartments where it reacts with O_2_, and, therefore, it is absolutely reasonable that in its fast flux towards the mitochondria through the mitochondrial membrane, NO-derived products would easily find ACO2 to be a readily accessible target given its known membrane proximity [60,61,62]. This results in ACO2 having a much lower K_i_ than the correspondent cytosolic enzyme, and also could explain the delay in timing and mechanism of inhibition of ACO2 vs. Complex and Complex II noted above, based on the lipophilic accessibility of proteins, complexes, and target clusters [47,62,63,64].

A year before these studies on tumor cells, Stuehr and Marletta had demonstrated that macrophages exposed to recombinant interferon-γ (rIFNγ) and LPS synthesize nitrite (NO_2_^−^) and nitrate (NO_3_^−^) [7,65]. Hibbs et al. put the pieces together and reported then that when murine macrophages become cytotoxic, they acquire the ability to convert l-arginine to l-citrulline, NO_2_^−^ and NO_3_^−^ through a pathway independent of the urea cycle [13,66]. In follow up work, Drapier et al. found a correlation between the ability of combinations of stimuli to induce NO_2_^−^ and NO_3_^−^ synthesis from l-arginine and their ability to cause reductions in aconitase activity in the producer macrophages [67]. There was also a similar pattern of inhibition of aconitase activity, Complex I and Complex II in macrophage effectors from *Bacillus Calmette Guerin* (BCG)-infected mice cultured with LPS in vitro. This course of inhibition in cytotoxic macrophages was similar to that measured in earlier studies in cytotoxic activated macrophages’ target cells; however, the kinetics of inhibition of Complex I and Complex II were more rapid and their residual enzymatic activity was lower [68]. In their experiments, indeed citrate dependent respiration (readout of aconitase and Complex I functionality) in cytotoxic macrophages began to decrease very early after stimulation while isocitrate oxidation (Complex I) had decreased to 30% only after few hours. Complex II activity (oxidation of succinate in the presence of rotenone), declined more slowly. This was also confirmed by Stadler et al. in NO-producing hepatocytes [69]. Finally, in 1990, experiments with Electron Paramagnetic Resonance spectroscopy in murine macrophages showed that the detection of nitrosyl-iron-sulfur complexes was dependent on the presence of l-Arginine in the culture medium. This work led to the conclusion that inhibition of mitochondrial non-heme iron-dependent enzymes in macrophages is mediated by autologous NO binding to enzymes containing iron-prosthetic groups [67,70].

These observations were not limited to traditional macrophages. Bolanos et al. found that astrocytes treated with LPS+FNγ had a marked decrease in their cytochrome c oxidase activity with no effect on Complex I or Complex II function and that this could be prevented by the specific NOS inhibitor N^G^-monomethyl-l-arginine (NMMA) (Figure 1a). These authors, however, were skeptical of which oxidant compounds might be responsible for this inhibition: they found that NO_2_^−^ + NO_3_^−^ released by LPS+FNγ treatment was accompanied by enhanced hydrogen peroxide (H_2_O_2_) production consistent with superoxide production [71]. Accordingly, scavenging of extracellular superoxide and peroxide by superoxide dismutase (SOD) and catalase (CAT) partially prevented both the increase of NO_2_^−^ + NO_3_^−^ species and the inhibition of the activities of these metabolic enzymes. They concluded that under their conditions, synthesis of peroxynitrite (ONOO^−^), a result of reaction between NO and superoxide [72,73,74,75] may be the culprit most likely through extracellular scavenging of NO. Since neither of these enzymes enters the cell, the O_2_^−^ and Fenton reaction-derived reactive oxygen species (ROS) were extracellularly preventing NO from diffusing from the petri dish. A study in rat skeletal muscle isolated mitochondria sheds light regarding the differences in enzyme inhibition seen in the various early works involving NO, nitrogen species, and mitochondrial function. Using GSNO in an effort to mimic NO-mediated injury, they described a defect in Complex IV concurrent with normal function of Complexes I to III. Inhibition of cytochrome c oxidase could be seen within minutes of adding the NO donor and ADP to mitochondria. These authors suggested that the use by many laboratories of digitonin to permeabilize cells might allow NO to be displaced and dissociate from Complex IV because of changes in intracellular osmoticity [37]. As a consequence, inhibition of Complex IV would not be observed. The inhibition of Complex I and II may be less labile, likely due to prolonged exposure. They suggested that the initial acute effect of NO on the mitochondrial ETC is the inhibition of respiration at cytochrome c oxidase, followed later in the stimulation by effects that involve the inhibition of Complexes I and II, either by NO itself or RNS [76] (Figure 2). Superoxide is generated within complexes I and III as byproduct of mitochondrial electron transfer which, if not consumed by mitochondrial manganese-dependent SOD (MnSOD) [77,78], can react with NO to form ONOO^−^ which is rapidly converted to NO_2_ and N_2_O_3_ [34]. Therefore, NO, by inhibiting cytochrome c oxidase, may contribute to the subsequent inhibition of Complexes I and II by promoting the generation of free radicals and possible RNS. Accordingly, inhibition of cytochrome c oxidase is required for the irreversible loss of Complex I [79]. Again different methodologies, in particular the use of whole cells rather than isolated mitochondrial fractions, likely account for differences in the various studies. Moreover, we should note that in vitro experiments are generally carried out at atmospheric O_2_ concentrations, which give rise to extracellular O_2_ concentrations of about 200 µM; way above what cells are exposed to in the body. This ‘hyperoxic’ state probably raises the concentration of O_2_ in mitochondria to non-physiological levels, providing resistance to endogenous NO [45].

All of this work is consistent with the observation that persistent inhibition of respiration by NO over a prolonged period of time will eventually result in the collapse of mitochondrial membrane potential, ATP depletion, and ultimately, cell death [80]. Behind these effects are several mechanisms like the fact that prolonged exposure of cells to NO can reduce the intracellular pool of glutathione through the formation of RNS [81,82,83] and render mitochondria more vulnerable to the deleterious effects of RNS. Although the effect of RNS depends on the composition of the microenvironment in which this substance would be generated [84], this situation facilitates the induction of the permeability transition pore (PTP), which leads to the collapse of the membrane potential, mitochondrial swelling, outer mitochondrial membrane rupture, release of mitochondrial pro-apoptotic factors, and cell death [85] (Figure 2b).

A peculiar property restricted to CI is what is called the active/deactive (A/D) transition, which can affect its susceptibility to modification, that may account upon persistent presence of NO or RNS. In brief, the enzyme spontaneously converts into the D-form after exposure to high physiological temperatures (>30 °C) or in the absence of substrate [86,87]. Deactive or dormant (D) enzyme is reactivated by a pulse of NADH, when the enzyme undergoes one or several slow turnovers or chemical conversions as NADH is oxidized and ubiquinone is reduced [87]. In highly metabolically active tissues, most of Complex I is in the A form. However, under conditions where the respiratory chain is in a reduced state due to the acute inhibition of cytochrome c oxidase and respiration rates are decreased, the catalytic efficiency of Complex I is slower because of the lack of the electron acceptor ubiquinone. As a result, the enzyme is converted into the D-form [88] (Figure 3).

One of the Complex I subunits, NDUFA9, and two of the mitochondrially encoded subunits, ND3 and ND1, are involved in the A/D conformational rearrangements. Interestingly, Cysteine^39^ in the ND3 subunit becomes exposed in the D-form and is susceptible to covalent modification by nitrosothiols (RSNO), ROS, and RNS [89,90,91], perhaps making it a specific site of physiological SNO regulation. Indeed, the A form of CI was found to be resistant while the D form is sensitive to inactivation by SNO. Re-activation of the D form by NADH was inhibited by nitrosothiols and peroxynitrite whilst the A form was insensitive to such treatment [92]. This mechanism was confirmed in human embryonic kidney cells (HEK) [88] where NO induced de-activation of Complex I in hypoxia leads to covalent modification of Cys-39 by NO-metabolites (Figure 3). Finally, in an intact heart model of cardiac ischemia/reperfusion Cys-39 of the ND3 was found nitrosated and responsible for inhibition of Complex I [93]. These unique qualities of CI provide an attractive hypothesis for the NO-dependent collapse of ETC complexes recently described (see below).

Taken together these early findings painted a clear picture suggesting the NO-mediated cytolytic mechanism of activated macrophages was feeding back to regulate the metabolism of these effector cells during activation. These observations of a collapse in mitochondrial activity represent a substantial feature in what is now referred to as metabolic reprogramming in macrophages. In fact, Drapier and Hibbs also described apparent compensating increases in glycolysis in stimulated macrophages. They concluded that inhibition of mitochondrial respiration by l-arginine-dependent effector factors results in a compensatory increase in activity of the glycolytic pathway and causes changes in the macrophage intracellular environment. Both of these effects would increase cellular resistance to certain facultative and obligate intracellular pathogens [68]. Published in 1988, this became one of the first descriptions of what is now referred to as “Glycolytic Commitment” of macrophages.

## 5. Macrophage Immunometabolism

Immunometabolism is a relatively new field driven by the idea that changes in metabolism govern the function of immune cells by controlling events of gene expression and protein synthesis [94]. For many years, researchers have tried to link inflammatory stimuli such as LPS to altered mitochondrial function [95,96], but recent technologies [97,98] have brought advances in the field. Krawczyk et al. [99] in 2010 published a landmark study in the field. They noticed similarities between signaling pathways activated by growth factors and those activated upon exposure to toll like receptors (TLR) ligands. Given that activation of signaling pathways downstream of growth factor receptors leads to altered cellular metabolism, it was hypothesized that similar metabolic changes would also be observed upon activation of dendritic cells (DCs, antigen-presenting innate immune cells) [100]. Since this time, the metabolic reprogramming of myelomonocytic cells has garnered rapidly growing interest in immunology. In response to LPS, DCs increase glycolysis and decrease respiration and oxidative phosphorylation (OXPHOS)-derived ATP [99]. This metabolic shift is also observed upon classic inflammatory activation with LPS+FNγ in BMDMs. Rodriguez-Prados et al. [101] demonstrated this at the level of gene expression and many groups [102,103] have now shown that metabolic changes can be present as early as 30 min after activation. The result is so called M1, proinflammatory macrophages, which demonstrate reduced mitochondrial activity together with substantial upregulation of glycolytic metabolism for the fast generation of ATP needed to sustain phagocytosis and microbial killing [102,104,105].

### 5.1. Current Model of M1 Macrophage Polarization

A pronounced feature of M1 polarized cells is what has been described as a “broken” mitochondrial TCA cycle. In brief, this interrupted TCA cycle is thought to drive the accumulation of key intermediates of metabolism such as citrate, itaconate, and succinate [106,107] that support effector functions. It has been suggested that downstream of LPS stimulation, the mRNA for the enzyme isocitrate dehydrogenase, IDH, is downregulated [106], facilitating the enlargement of a cellular pool of citrate to support the synthesis of fatty acids and the generation of lipid intermediates that promote inflammation (e.g., arachidonic acid-derived prostaglandins). Although reductions in mRNA do not always manifest in altered IDH protein abundance [108,109], stable isotope tracing confirmed a genuine break in the TCA cycle between citrate and α-ketoglutarate (α-KG) [106,110] consistent with reduced IDH activity. Decreases in IDH, it has been suggested, would also promote the accumulation of citrate-related metabolites that fuel the production of itaconate, which in turn, inhibits the succinate dehydrogenase complex (SDH) in mitochondria, leading to succinate accumulation [111,112].

Current models also suggest that in order to overcome the loss of carbon from the purported lack of IDH activity and the export of citrate form the mitochondria for cytosolic purposes, LPS-activated cells increase glutamine metabolism for the production of intermediates such as α-KG and succinate [102,106]. Surprisingly, although glutamine uptake and succinate levels increase, α-KG levels fall upon stimulation. These findings led to the suggestion that in polarized macrophages, pseudo-hypoxia, caused by reduction in α-KG /succinate ratios and suppression of dioxygenase activity [113,114], leads to HIF1α stabilization and expression of Pyruvate Dehydrogenase Kinase 1 (*Pdk1*) that phosphorylates Pyruvate Dehydrogenase (PDH), reducing pyruvate oxidation and promoting the Warburg effect [102,115,116,117,118]. Hence, the model predicts that macrophages primarily convert glucose-derived pyruvate to lactate, while anaplerotic glutaminolysis facilitates succinate accumulation, which supports HIF1α and IL-1β production and suppression of PDH via PDK1 [94].

Support for this framework model comes from multiple studies documenting profound increases in cellular citrate during M1 macrophage programming. Everts et al., from data in DCs showing strong incorporation of glucose carbon into citrate and high incorporation of ^14^C labeled glucose into lipid fractions in LPS-stimulated cells, favor a model in which glycolysis supports the de novo synthesis of fatty acids by fueling citrate early in activation. Their model proposes synthesis of fatty acids for the production of membranes needed for expansion of the subcellular endoplasmic reticulum (ER) and Golgi compartments to meet the demand for the increase synthesis and secretion of proteins upon stimulation [119]. This was first based on evidence that human histiocytoma cell lines where the mitochondrial citrate transporter had been silenced show deficits in prostaglandin mediators [120]. Later, decreases in PGE2 levels were found in DCs upon treatment with the fatty acid synthase (FAS) inhibitor C75 [119]. Therefore, the idea that citrate would support fatty acid synthesis leading to arachidonic acid, the precursor fatty acid for these inflammatory mediators, has been very much embraced in the field and set as a cornerstone of macrophage metabolic reprogramming.

Macrophage M1 polarization also results in profound extraction of carbon from the TCA in the form of itaconic acid. Itaconate is a rapidly produced metabolite generated by the decarboxylation of the TCA intermediate cis-aconitate. The reaction is carried out by Immunoresponsive Gene (*Irg-1*, a.k.a. aconitate decarboxylase 1; *acod1*). *Irg1* is one of the most highly upregulated genes during macrophage M1 polarization and although it has been near the top of many lists of upregulated genes in macrophages for years, it has only recently come into the forefront [107,121]. The biological role of macrophage-produced itaconate is currently the subject of much debate in the field. Itaconate is a potent suppressor of isocitrate lyase, the initiating enzyme of the glyoxlyate cycle used by bacteria when subjected to restricted nutrient conditions [122]. However, recent work has shown that itaconate plays a role in the generation of the elevated levels of succinate associated with macrophage stimulation. This is likely due to its weak, but substantial, inhibitory effect on SDH, although this mechanism is disputed [111,112]. Recent work has implicated itaconate in interferon responses, and inflammatory responses associated with *Mycobacterium tuberculosis* infection, Zika virus infection, and reperfusion injury of the heart or liver [123,124,125,126]. Itaconate production by the resident macrophages of the peritoneal cavity has also been shown to regulate their ROS production and facilitate their ability to promote tumor growth in this anatomical site [127]. Carbon tracing suggests contributions of glucose- but primarily glutamine-derived carbon during itaconate production [109]. The biology and biochemistry of itaconate has been recently reviewed in detail by others [128,129].

A third TCA-derived metabolite that accumulates during macrophage M1 stimulation is succinate. Succinate, derived primarily from glutamine-derived carbon accumulates due to the production of itaconate [102,112]. Succinate has been implicated in various aspects of inflammatory function including acting in trans via the succinate receptor expressed on internal organs as well as on immune cells [130]. As a fuel for Complex II, succinate can drive ROS production in macrophages and, therefore, be critical in antibacterial function [131], and via reverse electron transport, succinate can fuel ROS production by Complex I. Succinate and ROS have been shown to be involved, directly or indirectly, in the stabilization of HIF1α, which in turn drives expression of the inflammatory cytokine IL-1β [102,132]. A similar mechanism has been proposed for the involvement of succinate in damage associated with reperfusion injury in the heart [133]. Alternatively, succinate may regulate HIF1α via substrate inhibition of the dioxygenase family of enzymes including the prolyl-hydroxylases that regulate HIF1α accumulation [113,114].

In addition to accumulation of metabolites such as citrate, itaconate, and succinate, M1 metabolic polarization is associated with increased activity of the pentose phosphate pathway. This pathway is needed for production of nucleotides and NADPH, important for biosynthesis of lipids, production of effector ROS and RNS, and maintenance of intracellular redox status. The upregulation of the superoxide burst generated by the NADPH oxidase complex [134], together with the generation of reactive nitrogen intermediates through NOS2 [135], and increased production of effectors such as cationic peptides, lysosomal hydrolases, and metabolic inhibitors [136], including itaconate (antibacterial) [107] and 25-hydroxycholesterol (antiviral) [137], all contribute to macrophage antimicrobial responses. The importance of NO in effector function was further underscored by the finding that optimal NO production sustained by recycling of citrulline via argininosuccinate synthase and lyase (ASS1 and ASL) is fundamental for controlling mycobacterial infections [138]. Moreover, the NOS intermediate NOHA, supporting effective NO production by suppressing arginase activity freeing cellular arginine for NO production [139], has been shown to be critical in controlling parasites such as Leishmania [140].

In summary, current M1 polarization models suggest that rewiring of carbon utilization takes place through the concomitant suppression of IDH1 (via unknown mechanisms), SDH (via itaconate), and the PDH complex (via HIF1α– induced *Pdk1*) [102,106,117,119,141] and a subsequent increase in glutaminolysis. The resulting reduced respiratory rate could be necessary during inflammatory activation for (i) enhancing glycolytic flux to fuel cytoplasmic NADPH production and anabolism, (ii) generating ROS as redox signals, or (iii) increasing abundance of TCA cycle-associated signaling metabolites by breaking the TCA. 

As noted, the mechanisms responsible for the reduction in mitochondrial ATP and alterations in TCA function during M1 polarization can be multiple. It would seem clear from early work, however, that a contributing factor is likely to be the production of high concentrations of iNOS-derived NO during activation. In fact, various degrees of rescue of ATP-linked mitochondrial respiration can indeed be achieved upon genetic ablation or pharmacologic inhibition of NOS2 in myeloid cells [45,108,142]. Considering the history of NO chemistry and the elegant dissection of metabolic mechanisms associated with macrophage cytotoxicity noted above, it is surprising that most in the immunometabolism field have been speculating that the inhibitory effects of NO on oxidative mitochondrial metabolism might be simply due to direct inhibition of respiratory complexes, such as competitive inhibition of Complex IV by NO or s-nitrosation of Complex I [37,82,93]. Others have referred to evidence that many metabolic enzymes could be s-nitrosated by NO [143] and assumed that cysteine nitrosation could affect the activity of these enzymes in immune cells as well [144]. However, these NO effects were left hiding in plain sight as few were considering the degree to which NO might be directly involved in the metabolic switch that sustains function in activated innate immune cells. Recently, we and others, with the utilization of various NO donors and NOS2 inhibitors, have now revealed profound effects of NO suggesting important alternatives to the proposed models of macrophage metabolism [32,108,109]. 

### 5.2. Revised Model

In the last few years, new evidence necessitates that the field adapt a more comprehensive view on macrophage metabolism, requiring that NO be recognized as one of the central mediators of mitochondrial metabolic reprogramming during macrophage activation. Although, as noted above, NO had been suggested as a key intermediate in the metabolic switch of activated immune cells with nitrosation of cellular targets as the main proposed mechanism for its effects, the paucity of modern direct mechanistic studies led many authors to rely, in part, on metabolic principles extrapolated from other physiological systems that have led to over estimation of the role(s) of certain pathways in the establishment of metabolic phenotypes. 

#### 5.2.1. “Glycolytic Commitment”

The early dissection of macrophage cytotoxic mechanisms by Hibbs and Drapier, and the finding that the metabolic switch in macrophages and DCs activated with LPS was dependent on NOS2 led to the suggestion that NO generated by NOS2 mediates the LPS-induced mitochondrial dysfunction and increase in glycolysis [142]. The latter from data that showed that *Nos2*-deficient DCs do not increase their glycolysis in response to LPS. This has generated conflicting ideas in the field best explained by differences in cell type, NOS2 activity and timing within the response to stimuli at which the experimental procedures were performed. For example, Amiel commented [145] that in DCs that do not express *Nos2*, the rapid induction of glycolysis upon stimulation is present but transient, after which activated DCs return to a metabolic state closer to pre-activation. Indeed, the glycolytic burst upon TLR ligation leads to a rapid increase in glycolysis as reflected by extracellular acidification rates (ECAR) and this was found to be independent of NOS2. However, long-term glycolytic commitment, the term coined to indicate a state of full dependency of cell energy on glycolysis [142], is dependent on NO. This second wave of glycolytic induction is mediated in DCs by mTOR/ HIF1α /NOS2 [142,146]. This phenomenon has led many researchers to assume that HIF1α is critically required for DC glycolytic reprogramming, when in fact this is only the case for the sustained commitment to glycolysis observed in NO-producing DCs [145]. On this note, Kam et al. proposed a biphasic metabolic response to macrophage activation where both human PBMC-derived M1 macrophages and murine macrophage cell lines increase glycolytic rate after activation with LPS for as little as 40 min. This immediate elevation in glycolytic rate closely corresponds to M1 specific cytokine production (i.e., TNFα and IL-1β). In *Nos2* expressing cells, there is a subsequent, prolonged glycolytic response to activation [147]. This secondary upregulation of glycolysis requires the concurrent action of IFNγ with LPS and tightly corresponds to a decrease in mitochondrial respiration, depicted as oxygen consumption rate (OCR). This phenomenon starts at >4 h into stimulation. Increased NOS2 protein and activity were required for this secondary metabolic change.

Studies addressing macrophage membrane potential are also consistent with a prominent role for NO. Everts et al. found that mitochondrial membrane potential initially increased rapidly after stimulation with LPS [119]. Later, consistent with persistent NO-mediated reduction in mitochondrial OCR, mitochondrial potential decreased [99]. Rather than cellular efforts to balance mitochondrial proton gradients [131], these findings could be explained in further depth in light of NO chemistry. They align perfectly with a model where the hyperpolarization of mitochondrial membranes, the condition where the mitochondrial inner membrane becomes more negative on the matrix side [148,149], is induced by NO initially blocking Complex IV and favoring hydrolysis of glycolytically-derived ATP [80,150] (Figure 2a). Eventually, persistent inhibition of OXPHOS leads to further increases in glycolytic rates to guarantee sufficient supplies of ATP for ATPase. This model explains the second spike in ECAR values, a situation that although initiated by, may or may not be necessarily sustained over time by NO itself. 

#### 5.2.2. HIF1α -PDK Axis

Although the finding that NO is responsible for the suppression of OXPHOS associated with “glycolytic commitment” of macrophages was consistent with previous data in DCs, the work from Baseler et al. [151] also exposed apparent incongruences with studies indicating that HIF1α-dependent upregulation of *Pdk1* and consequent suppression of acetyl-CoA synthesis, may be responsible for OXPHOS regulation in macrophages [152]. With the finding of accelerated glycolysis in *Il10^−/−^* M1 macrophages which do not accumulate succinate, Baseler et al. argued against the proposed role for succinate in “pseudo hypoxia”-mediated HIF1α-PDK-1 activation. Although it is fairly recognized that HIF1α plays a role in enhancement of glycolysis, appreciated by increased expression of multiple HIF1α target glycolytic genes during LPS stimulation and their failed upregulation in *Hif1α^−/−^* background, they suggested that down-regulation of OXPHOS and subsequent commitment to glycolysis was mediated entirely via NO, and independent of HIF1α-driven *Pdk1* expression. 

By suppressing the production of NO, endogenous IL-10 preserves some OXPHOS in macrophages and disruption of this IL-10 rheostat system in knockout mice results in substantially increased NO production, increased glycolytic flux, and deeply suppressed OXPHOS. Together with the intact TCA break in *Il10* null cells, these data suggested a link between both aspects of macrophage metabolic programming, OXPHOX and TCA activity, where the presence of IL-10 favors a phenotype higher in respiration: all these effects mediated by via regulation of NO.

Another effect implicated in the diversion of glucose into lactate to support the glycolytic demands of polarized cells is the succinate-driven HIF1α induction of *Pdk1* and consequent phosphorylation and inactivation of PDH. We and others have detected this block using carbon tracing [109,153] and as a result of the lack of transition through PDH, the alternative route dependent on pyruvate carboxylation is evident, and the glucose contribution via acetyl-CoA to downstream metabolites is blunted. To our surprise, however, this phenomenon too is entirely dependent on NO while the HIF1α -PDK relationship is independent of NOS2. Although a role for *Pdk3* in this rewiring was found, Seim et al. concluded that that mechanism only partially reduces the flux through PDH [153]. Lastly, *Hif1α*
^-/-^ macrophages have been shown to maintain PDK1 expression, the stimulation-induced TCA break and to compromise PDH activity [101], and we found no significant effect of *Hif1α* deletion on LPS-induced glycolytic flux. In fact, this rewiring is apparently due to NO directly targeting PDH; more precisely, its dihydrolipoamide dehydrogenase (DLD) subunit (E3). This subunit is responsible for the dehydrogenation of the lipoate cofactor of the E2 subunit of PDH. In the process, it utilizes NAD^+^ as the final electron acceptor, generating NADH. We find that in M1 polarized macrophages DLD is directly inactivated, possibly by cysteine-nitrosation, in a NOS2-dependent manner. Older reports on the E3 of PDH support its targeting by nitroxyl (HNO) or peroxynitrite [154,155]. 

#### 5.2.3. TCA Break at Citrate

We revisited one of the most profound alterations in carbon flow in M1 stimulated macrophages, the break in the TCA between citrate and α-KG and the resultant decline in α-KG levels. Despite suggestions that this break was due to transcriptional down-regulation of *Idh1* by Jha et al. [106] we found that this phenomenon was the result of NO production. NOS2 is required for alterations in α-KG homeostasis and the lack of carbon transition between citrate and α-KG with subsequent accumulation of citrate, but it is dispensable for stimulation-dependent reductions in *Idh1* mRNA. Using cells lacking the NOS cofactor tetrahydrobiopterin, Bailey et al. also demonstrated the independence of the *Idh1* mRNA effect from NO, but noted an NO-dependent, stimulation induced, reduction in IDH activity [108]. However, while interrogating whole cell lysates, these assays are not specific for a given IDH (cytosolic or mitochondrial) and although others have shown decreases in IDH1 and 2 and justified concomitant reduced metabolic flux through oxidative and reductive IDH by measuring undistinguished isomers citrate/isocitrate in their experimental conditions [153], IDH1 is a cytoplasmic enzyme and, therefore, is very unlikely to directly affect TCA function. In contrast to IDH, several lines of evidence support that suppression of ACO2 is the source of the TCA break. First of all, as noted above, unlike IDH1, ACO2 is a long known target of NO via attack of its FeS cluster [53,68,69]. In addition, ACO2 protein levels and activity fall during macrophage polarization. Most importantly, during stimulation, citrate-mediated mitochondrial respiration is blunted whereas isocitrate respiration is intact. All these effects on ACO2 are dependent on NO [109]. 

#### 5.2.4. Lipid Accumulation

This refined model of macrophage metabolic rewiring subverts the proposed idea that citrate accumulation supports de novo lipid synthesis. The inability to generate NADH or FADH_2_ due to decreased activity of at least 3 enzymes involved in the TCA cycle slows respiratory chain activity and, as a result, OXPHOS. In concert, the ability of activated macrophages to oxidize fatty acids to CO_2_ is greatly reduced and the expression of transporters of fatty acids into the mitochondria for oxidation (CPT1 proteins) is also decreased. Multiple groups have found incorporation of fatty acids into triglycerides (TG) as a result. Accordingly, macrophage activation increases GPAT3 and DGAT2 gene expression and decreases TG lipolysis [156]. Importantly, a recent study sheds more light, revealing that the contribution of glucose to TG consisted only of the glycerol headgroups rather than in de novo synthesis of fatty acids [157]. These authors proposed that increased glycolytic rates serve to supply glycerol 3-phosphate for glycerolipid synthesis in activated macrophages. Specifically, neutral lipid accumulation in activated macrophages is not strictly dependent on increased uptake of fatty acids. Instead, it is the fate of internalized fatty acids that changes once macrophages are activated. Indeed, by relying on studies using inhibition of mitochondrial β-oxidation [158,159,160], Rosas-Ballina et al. proposed that impaired mitochondrial respiration could account for accumulation of neutral lipids in macrophages. Therefore, they tested if NO would contribute to this phenotype and found that indeed treatment with the NO donor DETA/NO in resting macrophages induced lipid droplets. This led them to the conclusion that iNOS-derived NO, by inhibiting mitochondrial respiration, and as a consequence fatty acid oxidation, leads to neutral lipid accumulation that is dependent on imported lipids. Such a model is consistent with findings that macrophage activation by ligands of TLR2 (zymosan), TLR3 (poly I:C), or TLR4 (LPS) all lead to TG accumulation in accordance with Somasundaram et al. and the ability to induce NO. In contrast, the low/medium NO flux levels elicited by cytokine stimulation would have minimal effects on lipid droplet accumulation [32,156] (Figure 4). We found that *Nos2^−/−^* BMDMs do not accumulate citrate and do not form lipid droplets, corroborating the hypothesis from Rosas-Ballina et al. that despite the widely emphasized idea that citrate export is the backbone of fatty acid synthesis, it in fact does not participate in TG accumulation. Consistent with our model, where glutamine is the major source of citrate during persistent LPS stimulation, data from ^13^C glutamine tracing showed that glutamine derived-citrate is not a relevant substrate for the lateral chains of newly synthesized fatty acids [157]. Lastly, we find that prostaglandin levels are maintained in *Nos2^−/−^* BMDMs further confirming the dispensable nature of excessive citrate in contributing to this pathway. While we argue against any role of citrate accumulation in the de novo synthesis of fatty acids, or as the basis for TG and lipid mediators of the inflammatory response, we do not reject the possibility that glucose-derived acetyl CoA, further oxidized into the proximal TCA cycle at citrate synthase, could contribute to some extent to de novo formation of phosphatidylcholine (PC) species fundamental in the early response to pro-inflammatory stimuli and are, therefore, NOS2 and NO independent. This may happen despite the apparent low expression of mitochondrial citrate exporter in murine macrophages as compared to human monocytes.

#### 5.2.5. Itaconate

One of the few TCA effects of metabolic reprogramming that is not entirely dependent on NO, is the rapid accumulation of itaconate. That is not to say that itaconate production is not affected by NO. Multiple labs have now documented accentuated itaconate production in macrophages lacking NO adding support to a NO-centric interpretation of reprogramming. The rapid and profound increase of itaconate in stimulated macrophages [112], *Nos2^−/−^* cells bearing higher levels of itaconate with high fractional incorporation of glucose carbon at the expense of glutamine carbon, together with the sensitivity of inhibition of PDH and ACO2 to NO, all justify a new, corrected model for this aspect of macrophage metabolic programming. This novel portrait of macrophage metabolism is now supported by data from multiple labs [32,108,109].

#### 5.2.6. Anaplerotic Pathways 

The shutdown of glucose utilization for mitochondrial OXPHOS in activated macrophages is accompanied with an influx of glutamine, presumably to provide carbon for the subsequent production of metabolites needed for the inflammatory response. Accordingly, NO regulates the extent to which compensatory pathways are exploited; glutaminolysis, pyruvate carboxylation, and malic enzymes are all routes that macrophages utilize more extensively during stimulation in an NO-dependent manner in order to replenish cellular intermediates important for anabolism [110]. Moreover, the fact that DLD is targeted and inactivated by NO has important implications because this E3 subunit of PDH is associated with other cellular metabolic enzymes, including oxoglutarate dehydrogenase (OGDH) and branched chain alpha keto acid dehydrogenase (BCKDH). This could be the basic reason why macrophages reportedly engage the GABA shunt, a metabolic pathway for the synthesis of succinate from glutamine-derived gamma-aminobutyric acid (GABA) [102]. Although no distinct mechanisms for engagement of the GABA shunt have been defined, and we find that many of the genes in the pathway are not substantially expressed in BMDM, in light of the new model of rewiring, it may represent an effort to bypass suppression of OGDH. Seim et al. directly assayed OGDH activity and found it to be profoundly suppressed in stimulated macrophages but attributed this phenomenon to changes in the lipoylation of the E2 subunit, which would also account partially as a mechanism for PDH suppression [153]. Although Tong et al. too had documented defects in the lipoate pool during macrophage activation [161], these effects, are NO-dependent and could be downstream of the direct targeting of the E3 by NO or RNS.

#### 5.2.7. ETC Complexes

A striking phenomenon that has not been fully described, and is often overlooked, in M1 macrophage polarization is the precipitous decline in the levels of mitochondrial ETC protein complexes late in stimulation. Some have suggested this might be a result of their post-translational modification based on older NO literature [144]. However, few studies have specifically investigated the biochemical basis of their disappearance or the impact of a lack of physical turnover in macrophages. We recently found that NO is sufficient, and iNOS is required for, this phenomenon. We feel the unique characteristics of Complex I described above provide insight to the likely mechanism. Although the mechanism of A/D transition may only operate in pathological conditions, when enzyme turnover is limited by hypoxia or by a high NO/O_2_ ratios [92,162], we suggested that it is the lack of substrates, (i.e., low NADH/NAD^+^ due to NO-mediated blunting of ACO2 and PDH enzymatic activities), that promotes the accumulation of Complex I in D form facilitating its modification and ultimately its degradation [163,164,165,166] (Figure 3). However, after 18h of stimulation, oxidation of isocitrate (Complex I) is still maintained indicating that inhibition of aconitase occurs before severe inhibition of the ETC complexes, absolutely in line with previous work [68]. Our idea is also supported by the fact that M1 macrophages when directly fueled in vitro after permeabilization can carry out Complex I and Complex II respiration, suggesting that even when NO concentrations have reduced ETC protein complexes, there are operational complexes mitochondria can rely on. Thus, one could speculate that spare and inoperative complexes are removed earlier in the process. Indeed, it is not surprising that blockade of NOS2 during stimulation rescues maximal respiratory capacity (MRC) and exposure to low NO affects MRC but not basal OCR [32,109,167].

Bailey et al. also found the collapse of the ETC to be NO-dependent but proposed an alternative mechanism from the Complex I point of view. In their model, in which tetrahydrobiopterin BH4 modulates specific metabolic changes in inflammatory macrophages by regulating NO production, respiratory function is regulated at an additional level. They report that NO synthesizing macrophages have decreased expression of *Ndufv2* but this was maintained in *Nos2^−/−^* cells. With this finding they have expanded the repertoire of NO-dependent mechanisms by which ETC is regulated to include governing the abundance of proteins making up the NADH-binding catalytic module of Complex I. Although they do not provide a mechanism, loss of this n-module subunit during stimulation would render Complex I incapable of binding and accepting electrons from NADH [108]. Although it is not known if either of these mechanisms provides the basis of the disappearance of Complex II as well, or if this is linked to decline of Complex I or suppression by itaconate, we propose that suppression of OXPHOS during acute NO exposure is due to substrate deprivation rather than via direct effects on mitochondrial ETC. It is intriguing how there does not seem to be effective replacement of these proteins leaving mechanisms open to further investigation. We have documented that many proteins involved in Fe-S cluster biogenesis, that are responsible for the assembly of the complexes, are affected by stimulation of macrophages [161]. Being targets of NO, the loss of these proteins during macrophage pro-inflammatory stimulation would impact the biogenesis of new functional respiratory enzymes.

#### 5.2.8. Metabolic Targets of NO Fluxes 

Although in vitro cultures of BMDM with subsequent treatment with conventional high doses of LPS+FNγ characterize a perfect model to study the effect of NO on cellular metabolism and these effects can be documented in vivo, NO is elicited at high levels under these conditions. Many studies have reported complex metabolic footprints in macrophages activated under different stimuli and tissue environments [168,169]. Therefore, macrophage mitochondrial rewiring during pro-inflammatory activation may not be universal and should not be generalized. IFNγ treatment alone, without LPS [170], or stimulation with monophospholipid A [171], or low LPS [112] does not result in the dramatic reductions in mitochondrial ATP production discussed above and in some circles maximal respiratory capacity (MRC) can actually increase early upon bacterial recognition [172]. Moreover, recall that Drapier had, some years ago, found correlations between the ability of various stimuli to elicit NO_2_^−^ and NO_3_^−^ production and their effects on aconitase [67]. Somasundaran et al. has now tied these different levels of NO production to different metabolic effects [32]. For example, they found that in higher density cultures, corresponding to higher extracellular NO flux, accumulation of citrate together with high citrulline levels were more appreciable. Only stimuli that generated medium to high NOS2-derived NO flux could affect OCR and strongly reduce OCR/ECAR ratios and this ability was, expectedly, proportional to cell density. Strong inhibition of OXPHOS required cellular NO ranging from 100–300 nM, a cellular flux 50–100 times higher than the physiologic levels involved in vascular tone [21]. Thus, only high NO flux induced by IFNγ+LPS induces full “commitment” to glycolysis. Moreover, intermediate levels of NO flux fail to promote citrate accumulation proving that this occurs only at relatively high NO concentrations [108,109]. This is in contrast to lactate production and broad glycolysis. Cytokines indeed can promote glycolysis independently of NOS2 as intracellular lactate levels are increased in BMDMs treated with low and medium NO inducers IFNγ+IL-1β and IFNγ+TNFα (Figure 4). Together, these data indicate that alterations in the density of cells, as well as the inflammatory environment affects local NO concentrations and further downstream metabolic effects [32]. The ability of varying NO fluxes to promotes distinct metabolic effects in macrophages opens intriguing questions in the field and should lead one to limit the extrapolation of all findings across all states of inflammation.

### 5.3. The Impact of NO-Mediated TCA Rewiring in Macrophage Function

Although our understanding of metabolic regulation of macrophages is expanding, the exact role of mitochondrial biochemical rewiring in inflammatory function during activation in murine BMDMs is not well understood. Although the early increase in glycolysis is important for DC and macrophage activation and is required for cell survival and cytokine production [99,102,151], macrophages appear to display a complete inflammatory phenotype despite the lack of mitochondrial metabolic rewiring. In our hands *Nos2^−/−^* macrophages even had elevated gene expression profiles and enhanced secretion of subsets of mediators of inflammation. Many studies in DC as well have reported an inhibitory role for NO on myeloid cell-mediated inflammation [142,145,146,173,174,175].

#### 5.3.1. Inflammatory Cytokine Production 

Although alterations in the TCA are considered to be vital for appropriate immune responses [115,176], evidence suggests that in cells that lack NO, the functional TCA cycle and stabilization of KEAP1 (Kelch-like ECH-associated protein 1), a key regulator of the transcription factor NRF2, prevents NRF2-mediated transcription. Our recent description of a NOS2-dependent NRF2 gene signature in murine macrophages, together with NRF2′s known ability to attenuate inflammation [177], suggests that NO reprograms metabolism while supporting the antioxidant response, all at the expense of full inflammatory potential. The data of Bailey et al. also support the conclusion that the inhibition of mitochondrial respiration is not essential for maintaining inflammatory polarization [108,178]. NO-deficient M1 macrophages still have elevated glycolytic rates and substantial levels of pro-inflammatory mediators TNFα, IL-6, and IL-1β. Both groups have reported elevated IL-1β in *Nos2^−/−^* macrophages likely due to the lack of NO-mediated nitrosation of the inflammasome complex [179,180,181]. Moreover, Bailey et al. observed increased transcription of *Il1b*, and although HIF1α can transcriptionally regulate *Il1b*, both labs concluded that HIF1α was not responsible for their findings.

#### 5.3.2. NRF2 Activation 

Many have come to the conclusion now that more likely, NO-dependent changes in NRF2 signaling could determine increased in pro-inflammatory gene transcription, like *Il1b* [109,182,183]. 

In contrast to a direct role for NO, recent work has suggested a direct metabolic connection to the activation of NRF2. Studies using derivatives of itaconate with unknown physiology and subcellular location, concluded that NRF2 responses in macrophages are elicited via itaconate [123]. The proposed mechanism involves itaconate-mediated alkylation of cysteine residues on KEAP1 promoting accumulation of NRF2 and transcription of target anti-inflammatory and anti-oxidant genes. However, we find that activation of NRF2 occurs regardless of intracellular availability of itaconate. In fact, activated *Nos2^−/−^* BMDMs accumulate higher levels of itaconate than WT cells and have comparable mitochondrial ROS yet they still lack substantial NRF2 activation [184]. The conclusion that itaconate is dispensable for NRF2 signaling has recently been confirmed by Sun et al. in a model of particulate-matter induced inflammation [185] and is consistent with McNeill et al., who found that activation-induced NRF2 in macrophages required Gch1-dependent NO production [183]. Lastly, Swain et al. have now corrected their previous conclusions by showing that although itaconate derivatives like dimethyl itaconate, 4-octyl itaconate or 4-monoethyl itaconate can affect NRF2 levels, these derivatives are not processed to itaconate correctly and real itaconate does not indeed perturb NRF2 [186]. Again here, the effects of NO have been hiding in plain sight as a plethora of older literature shows the direct ability of NO donors to drive expression of NRF2-controlled genes in multiple cell types [187,188,189,190,191,192]. Ongoing expression analysis and targeted metabolomics in BMDM from *Acod1^−/−^*, *Nos2^−/−^,* and double knockouts mice and the use of NO donors will likely definitively find that the NRF2 pathway is driven directly by NO as has been suggested for other cell types [193].

#### 5.3.3. Lipid Mediators of the Inflammatory Response 

Finally, although the accumulation of TG during pro-inflammatory stimulation discussed above may be important for macrophage ingestion and killing of microorganisms, or in sequestering and preventing their replication, the beneficial effects of lipid accumulation remain unclear. Lipid bodies also constitute the location for cytokines and enzymes involved in eicosanoid synthesis. Therefore, the NO-dependent increased formation of lipid vesicles during macrophage activation may facilitate the production and storage of compounds important for subsequent effector function [156,194]. Unlike the NOS2^−^ and NO-independent lipogenesis of species like phosphatidylcholine (PC), phosphatidylethanolamine (PE), and phosphatidylinositol (PI) which is important to support the need of cytokine secretion within the ER and Golgi network [195,196], the role of TG synthesis via DGAT1 is less clear. Although DGAT1-dependent TG synthesis has recently been reported to enhance macrophage inflammatory function together with PGE2 signaling [197], this finding still lacks mechanistic understanding regarding its intrinsic connections to metabolic needs and cell function. The fact that TG accumulation is NO-dependent while *Nos2^−/−^* cells have comparable PGE2 levels highlights the need for more research on this pathway.

### 5.4. NO and the Human Monocyte/Macrophage Dilemma 

Although human endothelial cells, hepatocytes, and other non-myeloid human cell types [198] have been shown to readily express NOS2 [199] and produce substantial fluxes of NO, many authors have reported limited induction of iNOS in human monocytes/macrophages causing continued controversy regarding the role of NO in human macrophage immunometabolism. It appears, for example, that while the murine *Nos2* locus has an open conformation, the human *Nos2* locus is plagued by hypermethylation in macrophages [200,201]. Moreover, even when there is clear evidence of detection of *Nos2* expression and resulting nitrotyrosine in human alveolar macrophages of chronically or acutely inflamed lungs [202,203,204,205,206,207,208], and exhaled NO can be measured from allergic asthma models of airway inflammation [209,210], in vitro the relevant cells fail to produce substantial levels of NO metabolites [211,212,213,214,215,216,217]. These observations have lead to suggestions of the presence of an endogenous inhibitor of NO synthase. An alternative hypothesis was that human monocytes/macrophages cannot synthesize tetrahydrobiopterin, an essential co-factor for iNOS activity and, thus, needed to acquire it from other cells, such as lymphocytes [218,219]. However, experimental evidence for these models is lacking [220]. Rather, it appears that highly specific conditions or special “priming” is required to elicit significant amount NO from human monocytes/macrophages. Growing evidence reveals substantial NO production by the human myeloid compartment under unexpected conditions that include the ligation of CD23, cross-linking of CD69, and IL-4 stimulation [221,222,223,224,225]. 

Considering the differences in NO fluxes in human and murine macrophages, it is interesting but not surprising that human peripheral blood monocyte (PBMC)-M1 macrophages maintain elevated glycolytic rates without undergoing a secondary phase as discussed above. This reflects lack of other bioenergetic effects of NO in human monocyte derived macrophages and is consistent with their “low output” NO production. For example, the mitochondrial bioenergetic changes observed in murine M1 polarized BMDMs are largely absent in similarly stimulated human PBMC-derived macrophages [100,226]. Accordingly, our older work, tracing carbon sources in M1 polarized human monocyte derived macrophages [227], showed full glucose-derived labeling of citrate and this pattern was conserved in downstream metabolites such as α-KG and malate. Clearly, glucose carbon flux is intact through the TCA in the absence of NO. 

Finally, by avoiding that autotoxic NO concentrations increase, human monocyte/macrophages may preserve a certain degree of metabolic and functional plasticity during M1 polarization, preserving mitochondrial bioenergetics to prevent collateral tissue damage and chronic inflammation [178]. It should not be a surprise that the immunosuppressive features of NO may have applied an evolutionary pressure to avoid the production of NO as a macrophage effector molecule. It seems to be a prime example of cost benefit balance.

These fundamental differences between human and murine models need to be exploited to help us understand the underlying biology. For example, the aforementioned association of NO production with IL-4 and IL-4 induced receptors such as CD23 suggests that M2 programming in human cells may employ NO to control glucose-derived carbon flux perhaps to facilitate the wound healing characteristics associated with glutamine metabolism.

## 6. Microenvironments and Niche Specificity

A great deal of our understanding of macrophage metabolic programming comes from in vitro study of bone marrow-derived cells. Given known metabolic plasticity in vivo and reports that redox states and metabolomes are altered during ex vivo manipulations [228], our in vivo study [109] of a modified Schwartzman-like reaction has important implications for the understanding of the regulation of immunity and disease pathogenesis. These data constitute one of the few investigations of macrophage TCA rewiring in vivo. Our findings showed that in vivo inflammatory macrophages do share many metabolic features with in vitro treated BMDMs. Although in our model peritoneal macrophages did not accumulate citrate, they displayed reduced levels of ACO2 activity and correspondent changes of citrate/isocitrate ratios demonstrating that the “break” in the TCA at aconitase is a relevant characteristic in vivo. Bailey et al. also used in vivo models of BCG infection and endotoxemia to demonstrate elevated levels of itaconate in lung tissues from leukocyte specific BH4-deficient animals, therefore, supporting once again NO-mediated regulation of itaconate in vitro and in vivo [108]. In addition to intracellular assessment, we studied the environment of peritoneal inflammation by measuring analytes in the lavage fluid. We detected alterations in the metabolic signature in the peritoneum with stimulation induced, NO-dependent, regulation of citrate, α-KG, arginine-derived metabolites, and itaconate. These data highlight that the macrophages not only respond metabolically to the nature of the surrounding environment but also at the same time alter that niche composition, enriching it with metabolites that possess immunomodulatory roles. This is particularly relevant in cancer. Considering the pro- and antitumor properties of NO [10,21], one would define tumoricidal macrophages in the tumor microenvironment (TME), as those that actively produce NO [229,230,231,232,233]. Somasundaram et al. suggest that higher densities of *Nos2* expressing cells leads to higher NO flux in the tissue that can impact morphology and oxygen consumption within the TME [32]. Of consequence, the increased available pO_2_ associated with elevated NO levels can affect the immune response [234]. Thus, in the microenvironment, the presence of NOS2 may regulate both the metabolism of the immune response and oxygen tensions, whereas the density of *Nos2* expressing cells may provide a fine-tuning mechanism for the extent of these effects. Together, the integration of these effects may define a powerful therapeutic target [235,236]. Therapeutically shifting the TME from tumorigenic simmering iNOS activity driven by the tumor itself, towards an immune surveillance situation where macrophages are reprogrammed to produce higher, effector level, NO fluxes could be beneficial [237,238]. 

## 7. Conclusions and Perspectives

We posit that the biochemical basis of macrophage metabolic programming should now been re-interpreted with the full awareness that this process is orchestrated by NO and that the mitochondrial rewiring is a result of, rather than a direct mediator of inflammatory polarization. The refined, NO-centric model suggests that during macrophage stimulation, there is a rapid increase in glucose uptake and rapid induction of *Irg1*. This results in synthesis of citrate and itaconate from glucose-derived carbon. *Nos2* is soon expressed and with the rise of intracellular NO levels PDH is directly suppressed reducing further utilization of glucose carbon in citrate and itaconate while diverting carbon to glucose-derived lactate. As NO levels continue to rise, ACO2 activity drops. Compensatory glutamine influx occurs at this point to make up for the lack of flux through PDH and the TCA [239]. This directs carbon through the TCA to support citrate, and itaconate production. Finally, itaconate production is limited due to NO suppression of ACO2 activity and accordingly glutamine-derived citrate accumulates. Lastly, as a result of substrate paucity Complex I reverts to the D form triggering the collapse of the electron transport apparatus and full glycolytic commitment (Figure 4).

With these recent studies, taken in conjunction with the pioneering early work on NO, our understanding of the metabolic reprograming associated with macrophage inflammation is substantially altered and new mechanisms are being defined as the role of NO moves into plain sight. Among these is the realization that this refined model suggests the intriguing possibility of NO-mediated metabolic regulation in trans. Not only are immune cells changing within but they are acting metabolically on neighboring immune cells, as well as nonimmune cells in proximity to the inflammatory response.

As a field, we must now reconsider a significant number of studies of myeloid cell metabolism such as landmark work depicting blockage of TCA cycle at IDH1 [106] and accumulation of certain metabolites with time during activation [153], and designate them as pre-“NO re-discovery”, re-interpreting them with the role of NO in mind. 

Lastly, it must be emphasized that the chemistry of NO, RNS, and their interactions with target thiols is complex and a product of the flux of NO, oxygen concentrations, ROS production, and other factors. Future work should embrace the chemistry of these entities toward unraveling the understanding of what proteins are affected by modifications, in what stoichiometry, and with what effect. Those of us working to understand metabolic programming in macrophages need to now delve into NO chemistry, including consideration of more reduced forms of NO such as HNO. HNO is produced by nucleophilic attack of a nearby thiol at the sulfur center of an s-nitrosothiol [240,241]. Compared to NO and other RNS, HNO has a distinct signaling mechanism of action [242,243] and higher electrophilicity than NO [244]. A principal modification mediated by HNO is the conversion of a thiol to the corresponding sulfinamide, RS(O)NH_2_, with a labile n-hydroxysulfenamide intermediate [240,244]. This is of particular note because, unlike -SNO, the generation of a sulfinamide in a biological system represents what is expected to be a much more stable thiol modification. Although it has yet to be demonstrated that sulfinamide formation from the reaction of HNO with thiols is physiologically relevant, this may be a mechanism of irreversible modification of metabolic activity. Indeed, the mechanism of irreversible inhibition of aldehyde dehydrogenase by HNO has been speculated to involve the presence of a sulfinamide [240,245].

Thus, although NO has been hiding in plain sight regarding its role in the regulation of macrophage metabolism, it has now emerged as a force to be reckoned with. The coming years will likely reveal additional impacts of this interesting soluble signaling molecule in metabolism and, as we gain a stronger appreciation of the potential impact of NO, HNO, and RNS in immune cell function, the scope and importance of addressing these effects will broaden.

## Figures and Tables

**Figure 1 metabolites-10-00429-f001:**
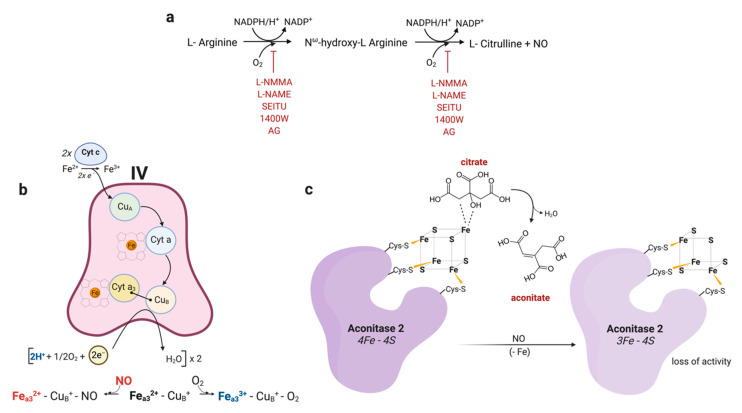
Biochemical mechanisms of Nitric Oxide synthesis and inhibition of mitochondrial targets. (**a**) Nitric oxide (NO) is synthesized from l-arginine by nitric oxide synthase (NOS) enzymes. The reaction involves the incorporation of oxygen (O_2_) into the unstable intermediate N omega-hydroxy-l-arginine and, subsequently, into l-citrulline, co-product of the generation of NO. N G -monomethyl-l-arginine (L-NMMA), N omega-Nitro-l-arginine (L-NAME), s-ethylisothiourea (SEITU), n-(3-(Aminomethyl)benzyl)acetamidine (1400W), and aminoguanidine (AG) are competitive inhibitor of NOS, the last two specific for inducible NOS (iNOS or NOS2), and block the pathway at the points indicated. (**b**) Nanomolar concentrations of NO immediately, specifically, and reversibly inhibit cytochrome c oxidase (Complex IV of mitochondrial electron transport chain, ETC) in competition with oxygen. NO reacts with the metals of the binuclear site of cytochrome c oxidase, binding tightly to the reduced iron in the distal pocket with the N closest to the Fe and the O close to, but not directly bonding, the CuB. (**c**) NO can target aconitase: NO is capable of oxidizing the cubane iron-sulfur (Fe-S) cluster that participates in the catalysis of the non-redox reaction of reversible isomerization of citrate to isocitrate with the intermediate cis-aconitate. This leads to iron release and consequent loss of the catalytic activity.

**Figure 2 metabolites-10-00429-f002:**
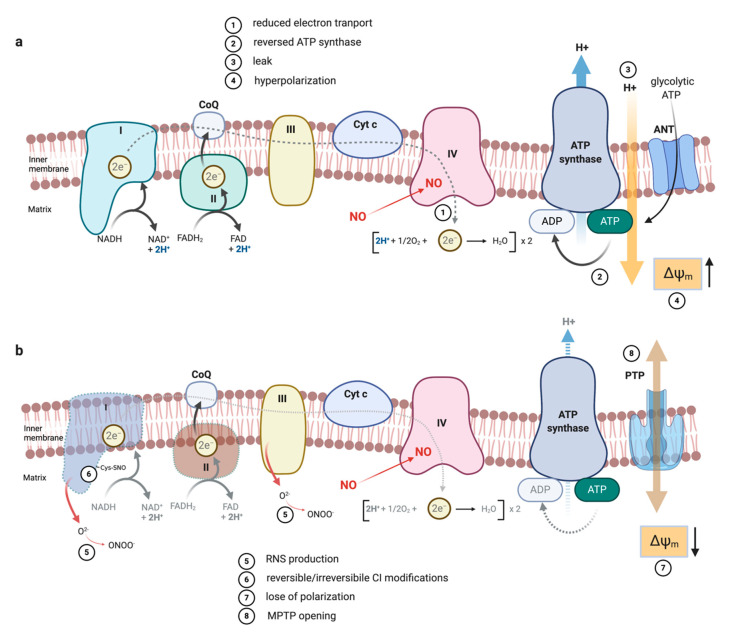
Mitochondrial bioenergetics following acute and prolonged NO injury. (**a**) Cytochrome c oxidase exert considerable control over the rate of respiration because NO increases the K_m_ of the enzyme for O_2_. Inhibition of cytochrome c oxidase by NO leads to a drop in the electron flux and consequently in proton extrusion. This changes the ETC to a more reduced state and leads to a defense response where reversal of the ATP synthase will hydrolyze cytoplasmic glycolytic ATP extruding protons (H^+^). This will re-establish the proton leak transiently lost during initial gradient dissipation. The effect of these changes is a hyperpolarization of the inner mitochondrial membrane. (**b**) Prolonged exposure to NO and, therefore, persistent inhibition of respiration by NO favors the formation of superoxide anions (O_2_^−^) and peroxynitrite (ONOO^–^), from the reaction between superoxide and NO, resulting from changes in intracellular redox conditions and depletion of the glutathione pool. These species can activate mechanisms that lead to irreversible damage, like deleterious modifications of CI. This situation facilitates the induction of the permeability transition pore (PTP), which leads to the collapse of the membrane potential and cell death. The passage of electrons (e^–^) along the ETC is depicted. Cyt c, cytochrome c; FADH_2_, flavin-adenine dinucleotide reduced; NADH, nicotinamide adenine dinucleotide reduced; Q, ubiquinone; I, II, III, and IV refer to the complexes of ETC. ANT, adenosine nucleotide translocator.

**Figure 3 metabolites-10-00429-f003:**
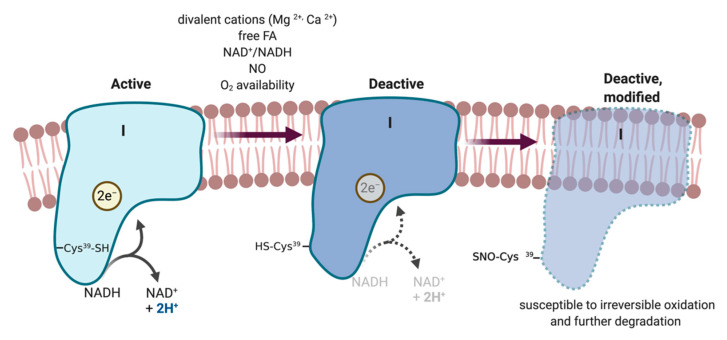
A/D transition of mitochondrial Complex I. The A-form of CI is capable of catalysis, but D-form is not. Divalent cations and free fatty acids influence the dynamics of the A/D transition. Majorly, substrate availability depicted by NAD^+^/NADH ratio in the matrix, together with presence of NO or hypoxic conditions strongly favor the kinetic of the A/D transition. Exposure of Cys^39^ of ND3 subunit of CI makes the D-form susceptible for thiol modifications that define deactivation reversibility, with nitrosation being the lead for further oxidation and potentially irreversible modifications and destruction of the enzyme.

**Figure 4 metabolites-10-00429-f004:**
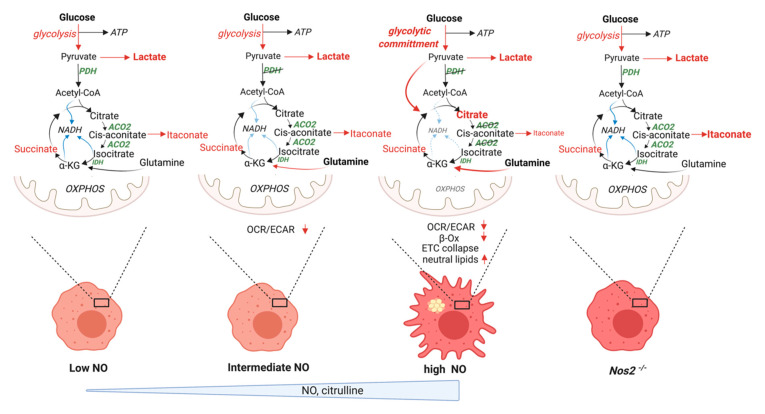
Orchestration of macrophage metabolism by NO. Macrophage metabolism switches to a high glycolytic status upon pro-inflammatory stimulation with increase lactate production, which is independent on NO and NOS2. Accumulation of metabolites like succinate and itaconate is a hallmark of pro-inflammatory macrophages. Treatment of Bone Marrow Derived Macrophages (BMDMs) with cytokines that elicit low NO fluxes (es. IFNγ+IL-1β) or the beginning of a strong signal does not result in further metabolic characteristics. With IFNγ+TNFα or cytokine combination that produce medium NO concentrations or with time after a strong stimulation, macrophages undergo oxidative shortage most likely from an interrupted flux through PDH, underlined by decreased oxygen consumption rate (OCR)/extracellular acidification rates (ECAR), with responsive compensatory anaplerotic pathways for TCA supply, like glutaminolysis. The presence of high concentration of NO in LPS+IFNγ-stimulated macrophages results in further multiple metabolic rewirings. Entrance of carbon into TCA via PDH is halted and ACO2 is inhibited. Compensatory carboxylation and glutaminolysis are enhanced, leading to accumulation of citrate and limitation in itaconate. The lack of NADH and reduced substrates generated by the broken TCA cycle leads to inactive mitochondrial complexes and suppressed OXPHOS, with significant appreciable drops in OCR and full commitment to glycolysis. In turn, β -oxidation of fatty acids (FA) is strongly decreased, leading to esterification of intracellular FA with glycolytic-derived glycerol for the synthesis of triglycerides and neutral lipids. In absence of NO (a situation represented by *Nos2^−/−^* LPS+IFNγ -stimulated BMDMs), higher glycolysis is maintained, PDH and ACO2 are intact and mitochondrial ETC displays full function. In this scenario, production of itaconate is strongly increased. The arrows represent the general direction of the metabolic flow in the system with the specific contribution of glucose and glutamine. The red color for arrows/ metabolites represents higher flux of depicted pathway/ increase in metabolite levels upon respective condition, compared to macrophage resting state. PDH pyruvate dehydrogenase; ACO2 mitochondrial aconitase; IDH isocitrate dehydrogenase; β -ox, β -oxidation; α-KG, α-KG, alpha-ketoglutarate; Acetyl-CoA, Acetyl-coenzyme A.

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
