# Peer review of "Nitric Oxide in Macrophage Immunometabolism: Hiding in Plain Sight"

_metabolites, 2020, doi:10.3390/metabo10110429_

Round 1

Reviewer 1 Report

This original review is devoted to the chemical properties and functions of Nitric Oxide (NO) and is focusing on the role of NO in macrophage reprogramming. Summerizing modern biochemical studies and recent publications the authors revised “mechanistic results of NO exposure” and proposed innovated model of metabolic rewiring macrophage, where NO play as master key of macrophages immunometabolism.

The review is undoubtedly interesting and valuable. The authors, keeping chronology, set out in details and systematically analyze the literary data, offering their own point of view.

Minor remarks.

The review is illustrated with wonderful diagrams, however, in my opinion, it is necessary to improve the visualization of captions (use a larger or brighter fonts), especially in Fig. 1 and 4.

Author Response

The review is undoubtedly interesting and valuable. The authors, keeping chronology, set out in

details and systematically analyze the literary data, offering their own point of view.

Response: Thank you for the comment that our manuscript could be interesting and relevant to the field.

Minor remarks.

The review is illustrated with wonderful diagrams, however, in my opinion, it is necessary to

improve the visualization of captions (use a larger or brighter fonts), especially in Fig. 1 and 4.

Response: We have modified the figures with increased fonts at the extent that was possible.

Reviewer 2 Report

This manuscript is a review article from the laboratories of two very senior investigators in the NO field. This group has investigated the mechanism of metabolic reprogramming in macrophages and has shown that proinflammatory molecules can lead to generation of M1 macrophages that are tumoricidal. They have shown that NO is intimately involved in regulation of inflammatory mediators and alteration in the electron transport chain.

The current review article is an excellent read for those who are entering this field and also for the seasoned investigators.

Author Response

The current review article is an excellent read for those who are entering this field and also for

the seasoned investigators.

Response: We thank the reviewer for the positive comments that the present review could be impactful for the field.

Reviewer 3 Report

The review “Nitric Oxide in Macrophage Immunometabolism: 2 Hiding in Plain Sight” by Palmieri et al. highlight the importance of NO in regulating macrophage metabolism that in turn is a central hallmark of their innate immune functions. The review correctly and strongly argues on acknowledging the role of NO as the major regulator of immuno-metabolic studies employing macrophages. The review encompasses work done in the NO field very thoroughly and nicely discusses how the field evolved from its discovery as a vasodilator to being a key regulator of TCA and OXPHOS in macrophages. The review also discusses recent work and proposes a model to resolve many discrepancies that exist in the field. However, there are few points that can improve the review than its current form.   

Major Comments:

  1. In many sections the authors argue for a revised model of macrophage polarization that places NO at the center. I feel that they have strong support for revising the model but the tone of the review is somewhat aggressive and could be moderated. For example, the authors state on line 456: “new evidence necessitates that the field adapt our view”…why not a new view? Or a more balanced view?
  2. The authors devote a large amount of space to dismissing or re-interpreting the data of others pointing out where it has failed to account for the effects of NO. They may be correct in this regard, but the text often deteriorates into a litany of finding fault with other studies.  I would much prefer that the authors simply state their case for a NO centric model and leave the conflicts they have with prior work out.  This would substantially shorten the text and clarify the arguments for the general readership.

  1. The studies are based primarily on the murine system and the authors do not highlight on how metabolic programming gets regulated in human macrophages that either lack evidence for, or produce little NO upon M1 inducing signals. It would be beneficial to discuss if these pathways are likley to be conserved (and if so what evidence is available vs. lacking) or if different mechanism predominate in human cells.
  2. Certain sections can be shortened with removal of unnecessary experimental details of papers discussed. For example, including results at different time points and intervals without providing an overview of why the timing is significant.
  3. The review also is confusing in some instances due to the use of terms and abbreviations that are difficult to comprehend unless one is highly familiar with NO biology. A glossary of terms would be helpful.
  4. The review could also be consolidated by grouping topics together rather than revisiting them several times in different sections (for example Hif1a-PDK).
  5. It would be helpful to add a figure that shows NO mediated regulation of key macrophage functions, metabolites, metabolic enzymes, transcription factors and ETC proteins.

Minor Comments:

  1. Use of punctuations in long sentences and their removal in short ones can help readers to follow the line of argument.
  2. Line 42, “macrophage lead…” can be changed to “macrophage produce…”.
  3. Line 69-70: The sole parasite reference can be moved to section pertaining to infections.
  4. Line 79-81: Can be shortened to “…. muscular relaxation, neuronal transmission and inhibition of platelet aggregation.”
  5. Line 85-90: A reference for these characteristics should be included.
  6. Line 94-97: This technical sentence can be removed. The sentence prior to this in Line 93-94 aptly summarize the findings of the paper discussed.
  7. Line 97-98: Can be changed to “They characterize levels of NO flux being low, intermediate or high respectively upon treatment with ….”.
  8. Line 91-108: The discussion of the paper can be curtailed and summarized in 3-4 major findings/sentences.
  9. Line 111-113: These sentences are difficult to comprehend and should be rephrased.
  10. Line 158-160: This sentence can be removed unless this particular time point holds any other significance from the point just mentioned above.
  11. Line 180-184: A reference suggesting this mode of NO action should be cited. It is not obvious to conclude this from text mentioned above or from the references in the sentence following this.
  12. Line 192-194: This sentence can be removed or appended with the next one with just stating the result without dwelling on motivation to test something or the other.
  13. Line 196-202: Discussion of this paper can be shortened to list important advances or differences from prior work.
  14. Line 204: “in victims of” can be replaced with “using” or “employing”.
  15. Line 228-229: This short result can either be removed or discussed with importance of late inhibition of complex II.
  16. Line 230-232: This sentence can also be removed or appended with the next one.
  17. Line 293-294: A reference should be added.
  18. Line 301: Describe NDUFA9, ND1 and ND1.
  19. Line 304-305: Define SNO.
  20. Line 350: A reference for linking rate of ATP generation and microbial killing or phagocytosis should be cited.
  21. Line 493-496: The authors might want to expand on “various reasons” and “continued controversy” regarding possible causes and effects of NO production in human monocytes/macrophages.
  22. Line 502: Define mitochondrial membrane hyperpolarization.
  23. Line 541-545: The authors should provide a reference if any which directly measure or show this flaw in the assays measuring citrate/isocitrate and IDH specificity.
  24. Line 550-556: The authors mention lower flux NO in human macrophages compare to murine macrophages to highlight the concept of NO induced TCA break. It would be nice to dedicate a small separate section on this difference, so that the readers make a note of this important aspect while going through the review.
  25. Line 603-604: Reference the papers that support this model or characteristics.
  26. Line 735: “…inhibits induces…”?
  27. Line 870: “Moreover, landmark studies such as Jha et al. work” can be replaced with summary of their findings.
  28. The last section of conclusion and perspectives can be shortened, rephrased and summarize the important points that can bring NO out of plain sight.
  29. Line 879-892: This can be shortened and moved to a place where nitrosylation and nitrosation are discussed or introduced for the first time.

Author Response

The review “Nitric Oxide in Macrophage Immunometabolism: 2 Hiding in Plain Sight” by Palmieri

et al. highlight the importance of NO in regulating macrophage metabolism that in turn is a central

hallmark of their innate immune functions. The review correctly and strongly argues on

acknowledging the role of NO as the major regulator of immuno-metabolic studies employing

macrophages. The review encompasses work done in the NO field very thoroughly and nicely

discusses how the field evolved from its discovery as a vasodilator to being a key regulator of

TCA and OXPHOS in macrophages. The review also discusses recent work and proposes a

model to resolve many discrepancies that exist in the field. However, there are few points that

can improve the review than its current form.

Response: we thank the reviewer for the constructive critique. We have tried to address the concerns as per the reviewer’s requests.

Major Comments:

  1. In many sections the authors argue for a revised model of macrophage polarization that

places NO at the center. I feel that they have strong support for revising the model but the

tone of the review is somewhat aggressive and could be moderated. For example, the

authors state on line 456: “new evidence necessitates that the field adapt our view”…why not

a new view? Or a more balanced view?

Response: Agreed, we have now smoothed our comments on other researchers’ work and proposed that the field would take a more “comprehensive” view of immunometabolism. We apologize for our overzealous approach in the first draft.

  1. The authors devote a large amount of space to dismissing or re-interpreting the data of others

pointing out where it has failed to account for the effects of NO. They may be correct in this

regard, but the text often deteriorates into a litany of finding fault with other studies. I would

much prefer that the authors simply state their case for a NO centric model and leave the

conflicts they have with prior work out. This would substantially shorten the text and clarify

the arguments for the general readership.

Response: Agreed, in the revised manuscript we have moderated our descriptions of previous work and deleted repeated and overly-dismissive comments on other researchers’ work.

  1. The studies are based primarily on the murine system and the authors do not highlight on

how metabolic programming gets regulated in human macrophages that either lack evidence

for, or produce little NO upon M1 inducing signals. It would be beneficial to discuss if these

pathways are likley to be conserved (and if so what evidence is available vs. lacking) or if

different mechanism predominate in human cells.

Response: We thank the reviewer for an excellent suggestion.  In response we have now included a dedicated paragraph for the discussion of the controversies on Nitric Oxide production in human macrophages and related findings and comments on their metabolic characteristics. (New lines 808-847; please be aware that the number of the lines correspond to “no markup” review mode of the word file)

  1. Certain sections can be shortened with removal of unnecessary experimental details of

papers discussed. For example, including results at different time points and intervals without

providing an overview of why the timing is significant.

Response: Agreed, in the revised manuscript we have deleted excessive details on experimental time points and conditions.

  1. The review also is confusing in some instances due to the use of terms and abbreviations that

are difficult to comprehend unless one is highly familiar with NO biology. A glossary of terms

would be helpful.

Response: Agreed, we have now added all the abbreviations of NO chemistry and biology and immunometabolism mentioned in the main manuscripts in a dedicated space after the abstract.

  1. The review could also be consolidated by grouping topics together rather than revisiting them several times in different sections (for example Hif1a-PDK).

Response: We appreciate this suggestion.  We have now addressed this point by grouping topics in sub-paragraphs related to the section of the “revised model” of immunometabolism. Accordingly, to make the topics‘ order organically organized to allow a logical and connected flow of the discussions, we moved the order of some paragraphs from the original layout of the first submitted manuscript.

  1. It would be helpful to add a figure that shows NO mediated regulation of key macrophage

functions, metabolites, metabolic enzymes, transcription factors and ETC proteins.

Response: Agreed, we believe the graphical abstract provided would be extremely helpful for the readership to understand the key points of NO regulation of macrophages.

Minor Comments:

  1. Use of punctuations in long sentences and their removal in short ones can help readers to

follow the line of argument.

Response: Apologies. We have reviewed the text and tried our best to make the flow of the text more fluid.

  1. Line 42, “macrophage lead…” can be changed to “macrophage produce…”.

Response: We have corrected the text according to the suggestions. (New line 66)

  1. Line 69-70: The sole parasite reference can be moved to section pertaining to infections.

Response: In the reorganized text, we have now moved this reference to the section of the “current model of of M1 Macrophage Polarization”, including it within the importance of NO in effector function. (New lines 454-456)

  1. Line 79-81: Can be shortened to “…. muscular relaxation, neuronal transmission and

inhibition of platelet aggregation.”

Response: Agreed, we deleted part of the text according to the reviewer’s suggestions.(New lines 101-102)

  1. Line 85-90: A reference for these characteristics should be included.

Response: Agreed, we have now added references in support of the importance of the sustained and high levels of NO in immunity as opposed to nanomolar pulses necessary for cardiovascular responses. (New line 111)

  1. Line 94-97: This technical sentence can be removed. The sentence prior to this in Line 93-94

aptly summarize the findings of the paper discussed.

Response: We have removed unnecessary experimental details as suggested.(New modified paragraph starting at line 112)

  1. Line 97-98: Can be changed to “They characterize levels of NO flux being low, intermediate or

high respectively upon treatment with ….”.

Response: Thanks, we have modified the text accordingly. (New lines 116-117)

  1. Line 91-108: The discussion of the paper can be curtailed and summarized in 3-4 major

findings/sentences.

Response: Agreed, we shortened the comments on this paper as suggested. (New lines 112-124)

  1. Line 111-113: These sentences are difficult to comprehend and should be rephrased.

Response: Agreed, we have modified the text accordingly. (New lines 127-130)

  1. Line 158-160: This sentence can be removed unless this particular time point holds any other significance from the point just mentioned above.

Response: We have deleted this sentence as suggested. (New lines 184-186)

  1. Line 180-184: A reference suggesting this mode of NO action should be cited. It is not

obvious to conclude this from text mentioned above or from the references in the sentence

following this.

Response: We have now added the reference specifically corresponding to the statements at these lines as suggested. (New line 210)

  1. Line 192-194: This sentence can be removed or appended with the next one with just stating the result without dwelling on motivation to test something or the other.

Response: We have deleted this sentence as suggested. (New lines 218-219)

  1. Line 196-202: Discussion of this paper can be shortened to list important advances or

differences from prior work.

Response: Agreed, as suggested we have now shortened this part by highlighting first the relevance of this work and deleting some experimental details. (New lines 221-225)

  1. Line 204: “in victims of” can be replaced with “using” or “employing”.

Response: We have modified this as suggested. To avoid a misunderstanding, we substituted the word “victim” with “target”. (New line 225)

  1. Line 228-229: This short result can either be removed or discussed with importance of late

inhibition of complex II.

Response: We have deleted this result as suggested. (New lines 251-254)

  1. Line 230-232: This sentence can also be removed or appended with the next one.

Response: We have now linked this statement to the following phrase as suggested by the reviewer. (New lines 254-257)

  1. Line 293-294: A reference should be added.

Response: We have added the reference corresponding to these findings as suggested. (New line 316)

  1. Line 301: Describe NDUFA9, ND1 and ND1.

Response: A statement of description of these subunits has now been added correcting this oversight. (New lines 323-324)

  1. Line 304-305: Define SNO.

Response: A definition has now been added. (New line 326)

  1. Line 350: A reference for linking rate of ATP generation and microbial killing or phagocytosis

should be cited.

Response: As suggested, two references in support of this phrases have now been added. (New line 373)

  1. Line 493-496: The authors might want to expand on “various reasons” and “continued

controversy” regarding possible causes and effects of NO production in human

monocytes/macrophages.

Response: As stated in the response to the reviewer’s major point #3 we have now provided a specific paragraph for findings in human monocytes/macrophages. (New lines 808-847)

  1. Line 502: Define mitochondrial membrane hyperpolarization.

Response: We have now added a definition and relative reference for mitochondrial hyperpolarization. (New lines 523-524)

  1. Line 541-545: The authors should provide a reference if any which directly measure or show

this flaw in the assays measuring citrate/isocitrate and IDH specificity.

Response: We have now added brief statements clarifying how the detection of citrate/isocitrate and IDH specificity for the mentioned studies may lack some accuracy. (new lines 574-577)

  1. Line 550-556: The authors mention lower flux NO in human macrophages compare to murine macrophages to highlight the concept of NO induced TCA break. It would be nice to dedicate a small separate section on this difference, so that the readers make a note of this important aspect while going through the review.

Response: Noted.  Please see response to major point #3 and minor point #21.

  1. Line 603-604: Reference the papers that support this model or characteristics.

Response: We have now added the references that reinforce the new model of immunometabolism. (New line 633)

  1. Line 735: “…inhibits induces…”?

Response: We have corrected this typo. (New line 737)

  1. Line 870: “Moreover, landmark studies such as Jha et al. work” can be replaced with

summary of their findings.

Response: We have now modified this section by summarizing the findings of the works cited. (New lines 902-903)

  1. The last section of conclusion and perspectives can be shortened, rephrased and summarize

the important points that can bring NO out of plain sight.

Response: Based on the reviewers suggestion, we have modified this section substantially. (New lines 880-926)

  1. Line 879-892: This can be shortened and moved to a place where nitrosylation and

nitrosation are discussed or introduced for the first time.

Response: Excellent suggestion.  We have now moved these definitions of nitrosylation and nitrosation to the end of paragraph 2 (“production”) where nitrosative stress and modifications are introduced. (New lines 135-145)

Round 2

Reviewer 3 Report

The authors have made many improvements in the revised version and I am now convinced it provides a balanced and comprehensive review of the topic.